# Development of a Cat Behaviour Issues Assessment Scale (CABIAS) Assessing Problem Behaviours in Cats

**DOI:** 10.3390/ani13182992

**Published:** 2023-09-21

**Authors:** Kevin McPeake, Andrew Sparkes, Charlotte Billy, Sarah Endersby, Jean François Collin, Xavier De Jaeger

**Affiliations:** 1The Royal (Dick) School of Veterinary Studies and The Roslin Institute, Easter Bush Campus, Midlothian EH25 9RG, UK; kevin.mcpeake@ed.ac.uk; 2Simply Feline Veterinary Consultancy, Shaftesbury SP7 8FY, UK; andy@sparkes.uk.net; 3Ceva Santé Animale, 10 Av. de la Ballastière, 33500 Libourne, France; charlotte.billy@ceva.com (C.B.); sarah.endersby@ceva.com (S.E.); jean-francois.collin@ceva.com (J.F.C.)

**Keywords:** pheromone, cat, feline, scale, FELIWAY, scratching, urine marking, fear, cohabitation, index score, problem behaviours

## Abstract

**Simple Summary:**

Problem behaviours in cats, such as scratching, urine marking, inter-cat cohabitation issues, and fear, are a welfare problem for both cat and caregiver. The aim of this study was to develop a new Cat Behaviour Issues Assessment Scale (CABIAS) to assess and monitor these behaviours in cats. The validity and reliability testing of the CABIAS undertaken in this study demonstrated that it could be a valuable tool for the assessment of these problem behaviours in cats. The level of agreement between different caregivers assessing the same cat was excellent. This CABIAS might be a useful tool for caregivers to assess and monitor these problem behaviours in cats. The CABIAS could also be used as a clinical tool by veterinarians and behaviourists when assessing problem behaviours in cats and monitoring responses to treatment.

**Abstract:**

Problem behaviours in cats, such as urine marking, scratching, fear, and problems of cohabitation between cats, can present a potential welfare problem for those affected cats and have a negative impact on the human–animal bond. The aim of this study was to develop a Cat Behaviour Issues Assessment Scale (CABIAS) for caregivers to assess these common problem behaviours in cats, and to investigate aspects of the validity and reliability of the CABIAS. The CABIAS uses an index score as a scoring system combining aspects of the frequency and intensity of the problem behaviour. An electronic survey was developed, and 384 households, each with two participants able to observe the cat’s behaviour, were recruited for the study. The participants were asked to record their cat’s behaviour independently at weekly intervals over a 6-week period. A FELIWAY Optimum diffuser (containing a commercial feline pheromone complex) was provided to half of the participants, to be used during part of the study. The participants were subdivided into four groups according to whether a problem behaviour was present (or not) and whether the pheromone diffuser was provided (or not). The results demonstrated that the index scores for each problem behaviour had very high inter-rater reliability. There was a high correlation between the index scores and the impact the problem behaviours had on the caregivers. In those cats with a declared problem behaviour where the product was used, a reduction in index scores was observed over the duration of the study. The CABIAS was shown to exhibit satisfactory validity, reliability, and sensitivity to change in the current study, suggesting that it may be valuable as a tool for assessing and monitoring scratching, inter-cat cohabitation, urine marking, and fear in cats. The CABIAS could be used by veterinarians and behaviourists to assess these problem behaviours in individual cats and monitor responses to treatment. The CABIAS could also be considered as a research tool to assess the efficacy of interventions aimed at improving these problems.

## 1. Introduction

Urine marking, scratching, fear, and problems with cohabitation between cats are amongst the most common behavioural problems reported by caregivers [1,2,3,4]. These problems are often associated with negative emotional states such as fear, anxiety, and frustration, and they can present a welfare problem for the cat [5,6]. They also have a negative impact on the human–animal bond, potentially resulting in euthanasia or relinquishment of the cat [7]. Assessing problem behaviours is thus a crucial aspect of feline health and welfare [8,9], although the complexity of feline behaviour can make accurate assessment and diagnosis of these problems challenging.

When evaluating such problems, there is a need for scales that can objectively measure the behaviours of interest, avoiding subjective assessments and terms which are challenging to standardise [10]. Various questionnaires and rating scales have been developed for assessing feline personality [11,12], and recently as tools to collect quantitative data on normal and problematic behaviour in cats [12,13]. Currently available scales for assessing and monitoring cat behaviour are focused on traits (personality) or have categories that offer a less precise assessment of the intensity or frequency of problem behaviours for individual cats. Therefore, there is a need for a standardised and validated tool that can be completed by caregivers and used by veterinarians and behaviourists to precisely record the frequency and severity of problem behaviours. This allows accurate data to be collected at the time of assessment, and the same measure can be repeated at intervals to assess responses to treatment.

The objective of this study was therefore to develop a scale that is usable by caregivers for assessing four specific common problem behaviours in cats, and to assess aspects of the validity and reliability of the Cat Behaviour Issues Assessment Scale (CABIAS). The development of the CABIAS was designed to improve the diagnosis and monitoring of these problem behaviours in cats, leading to better feline health and welfare.

## 2. Materials and Methods

### 2.1. Selection

The participants were part of a French panel of cat caregivers and were recruited by the marketing company “IMASENS”. During the selection process, cats owned by the participants were classified into five groups based on the main problem behaviour that concerned the caregiver (i.e., the primary problem), or no problem. For cats with multiple problem behaviours, the primary problem determined the group to which the cat was assigned. The five groups were as follows:No reported problem behaviour.Problem scratching—defined as “scratching on vertical surfaces indoors other than on a scratching post (e.g., a sofa, carpet, curtains, or furniture)”.Problem urine marking—defined as depositing urine in a standing position, indoors, and against a vertical surface outside the litter box.Problem fear—defined as hiding or running away.Problem cohabitation—defined as difficulty cohabitating with other cats in the same household (e.g., fighting, biting, conflict, crying, chasing, staring, hissing, blocking etc.)

Selection continued until there was a minimum of 80 cats in each of the groups, although many cats exhibited more than one problem. A total of 444 households were recruited to the study after having read and accepted the letter of information and consent received by email after selection.

The study was reviewed and approved by the Ceva Santé Animale Ethical Committee (CFAEC-2021-23 and CFAEC-2021-24) and by the Veterinary Ethical Review Committee of the Royal (Dick) School of Veterinary Studies, University of Edinburgh (VR 46.22).

### 2.2. Inclusion and Exclusion Criteria

To be eligible for inclusion in the study, there had to be one or two cats ≥12 months of age in the household, and at least one of the cats had to be exhibiting one of the four specified problem behaviours, or the cat(s) had to be free of any of the four problem behaviours considered. Households needed to have at least two caregivers aged 18 or older who were able to observe the cat during the study period and who provided informed consent to participate to the study. The first caregiver to register to the panel was considered to be the principal caregiver. The exclusion criteria are listed in Table 1, and post-inclusion removal criteria (for the removal of a participant after starting the study) are shown in Table 2.

### 2.3. Study Protocol and Data Acquisition

Caregivers were asked to complete online web-based questionnaires designed to assess their cat’s behaviour. During the study, the two caregivers were asked to independently rate their cat’s behaviour and to refrain from communicating their answers to one another; they were also asked to avoid any environmental changes for the cat over the course of the study period. The caregivers were notified via email when each questionnaire was available and were given 48 h to complete it (with a reminder sent after 24 h where necessary). Regardless of whether the caregivers had declared a problem behaviour or not at the start of the study, they were instructed to document specific aspects of their cat’s behaviour on a weekly basis.

The study period lasted 6 weeks, and behaviour questionnaires were completed at weekly intervals (Figure 1), giving a total of 7 questionnaires (including day 0). In addition, general questionnaires were also completed every week, covering aspects of the environment, whether caregivers and cats had been following the study protocol, etc.

During the first two weeks (Phase 1), there was no intervention in any of the households, and this period served as a baseline for behaviour assessments. From day 14 (Phase 2), 186 households that had been chosen during the selection process from those households where caregivers had declared a problem behaviour were asked to use a commercial pheromone diffuser (FELIWAY Optimum diffuser, Ceva Sante Animale, Libourne, France), and the caregivers were asked to plug the diffuser into an electrical socket in the room where the cat(s) spent most of their time. The other caregivers did not use anything during the Phase 2. All caregivers, whether allocated a diffuser or not, were asked to continue completing weekly behaviour questionnaires for their cat(s) for the remaining four weeks (Figure 1). At the end of the study, a gift card for EUR 30 was provided to each caregiver.

Each behaviour questionnaire asked caregivers to assess the frequency and intensity of the four problem behaviours of interest (irrespective of whether they had declared that the cat was exhibiting a problem behaviour) during the preceding week. Frequency was reported on a semi-quantitative 7-point scale (0 = never; 1 = once a week; 2 = twice a week; 3 = every 2 days; 4 = almost every day; 5 = every day, once or twice a day; 6 = every day, more than twice a day). The intensity of the behavioural problem was reported on a visual analogue scale (VAS) from 1 to 10 (with intervals of 0.1). The caregivers were specifically asked to ignore frequency when reporting the intensity. If a caregiver responded “Never” to the frequency question, the question on intensity was not asked. Where intensity was scored, the scale had a minimum value of 1 (to avoid an index score having a value of “0” other than if the frequency reported was “Never”). To quantify the problem behaviour, an index score was calculated as the product of the frequency multiplied by the intensity (with results ranging from 0 to 60). Multiplying frequency by intensity is also used in pain and behaviour assessment [14,15,16,17,18,19], and a similar index score was used for those behaviours in a previous study on the use of FELIWAY Optimum [20].

The caregivers were also asked each week whether they considered the specific behaviour to have been a problem during previous seven days (yes/no). Additionally, the caregivers were asked to indicate the extent to which the behaviour was impacting their lives using a VAS of 0–10 (from 0 “it is not impactful at all” to 10 “it is extremely impactful”).

### 2.4. Statistical Analysis

Descriptive statistics were used for demographic data. For each problem behaviour, the index score was calculated and analysed alongside the caregiver’s declaration about whether they perceived a problem to be present or not, and the rating of the impact of the problem on the caregiver’s life.

The following analyses were performed to assess aspects of reliability and validity for each of the problem behaviours:

To assess the association between results from caregivers within each household, pairs of index scores were analysed using the intraclass correlation coefficient (ICC). A previous study has suggested that that ICC values <0.5 indicate poor reliability, values of 0.5–0.75 indicate moderate reliability, values of 0.75–0.9 indicate good reliability, and values >0.9 indicate excellent reliability [21].

Additionally, to assess the agreement between the two caregivers, their index score difference was quantified and plotted. The percentage of observations where the difference between the two index scores (range 0–60) was ≤10 was calculated. For all other analyses, only the principal caregivers’ values were used.

Convergent validity was assessed through correlational analysis of the index score and the VAS of the impact of the behaviour on the caregiver. It was hypothesised that the higher the index score, the higher the impact on the caregiver would be. The index score is a product between a Likert-like score (frequency) and intensity, so this score is a continuous parameter. Thus, a Pearson correlation coefficient between these values was calculated for each timepoint and for each problem behaviour.

To investigate the responsiveness of the index score, a linear mixed model based on data collected during the second part of the study (days 14–42, when some of the households were using a pheromone diffuser) was used. The change in index score from D14 was used as the response variable, with day, treatment, and their interaction as fixed covariates and the cat identifier as a random effect. The slopes of the estimated regression lines of the index score change from D14 were calculated for each group; these values provide an indication of the general evolution of the index score in the different groups.

The efficacy of the pheromone diffuser (FELIWAY Optimum) in reducing problem behaviours in cats has been previously reported [20]. The pheromone provides a reassuring message to the cats about some aspect of their environment, and this should help the cat to adapt its behaviour [22]. It was predicted that for cats with a declared problem behaviour, the index score would be reduced for the group exposed to the pheromone product but would not appreciably change where no pheromone product was being used.

Respondent-related validation was assessed by comparing index scores for the previous 7 days with whether the caregiver considered a problem behaviour to be present during the previous 7 days (yes/no answer). This comparison indicated the extent to which the index score was discriminant or not, meaning that lower index scores or scores equal to 0 should be associated with the answer “no”, and when the caregiver answers “yes” this should be associated with index scores superior to 0.

SAS software (Version 9.4) was used to provide the statistical results.

## 3. Results

### 3.1. Population Definition

Out of the 444 households initially included, 60 were removed after their inclusion in the study, for the reasons presented in Table 3. This left a total of 384 households that completed the study, representing 494 cats (274 single-cat households and 110 households with two cats). By definition, only households with two cats could exhibit cohabitation issues.

The mean age of the cats was 6.1 ± 3.8 years (median [Q1–Q3] = 5 [3,4,5,6,7,8,9]); 54% were female and 46% male, and 85% of the cats were neutered. No cat was removed due to any adverse event. The problem behaviours shown by the cats are shown in Table A1 and Table 4. As can be seen from Table A1, most cats presented more than one problem behaviour.

For data analysis and graphs, the cats were classified according to the problem behaviour(s) that they displayed (as documented by the owner at selection), as well as according to their product group (i.e., pheromone diffuser or no product). Thus, the number of cats per group differed for each behaviour (see Table 4).

### 3.2. Evaluation of Aspects of the Behavioural Scale (Index Score)

#### 3.2.1. Inter-Rater Reliability

Households were specifically recruited with two caregivers to assess the inter-rater reliability of the scale for the four behaviours. Table 5 shows the ICC values for the index scores for each problem behaviour at each timepoint, along with the mean for each problem behaviour. All means were >0.9, suggesting excellent inter-rater reliability [21] under the conditions of the study.

To assess the precision of agreement between the two caregivers, the difference in index scores between the two caregivers was assessed. This demonstrated a high proportion where the difference was close to 0 (with 0 representing perfect agreement; see Figure A1). In addition, the proportions where the difference was ≤10 are provided in Table 6 (80% to 95% for the four problem behaviours). These results demonstrated that the index scores provided by the two caregivers were generally close, meaning the numerical agreement between them was high.

#### 3.2.2. Aspects of Validity

To assess the convergent validity, Pearson’s correlation coefficients were calculated between the index score of a problem behaviour for the previous 7 days and the VAS score in answer to the question on the impact of the behaviour on the caregiver in the previous 7 days. These were calculated for each timepoint and for each behaviour (see Table 7).

The coefficients of correlation were again very high, with values close to 0.9 for all four behaviours, meaning the index scores converged well with another way to estimate the behaviour of the cat, based on how much it impacted the caregiver. The graphical representations, including all timepoints, are shown in Figure A2.

For the responsiveness and respondent-related aspects of the validity of the index scores, the four behaviours are presented independently.

##### Scratching

First, the percentage of caregivers declaring that scratching was considered to be a problem during the previous 7 days is presented in Figure 2A. This demonstrates that at D0 the consideration of the problem was consistent with the reported incidence at selection, aside from the delay between the two questionnaires (1–2 weeks). This figure also shows that the caregivers’ declarations were consistent over 14 days when no product was applied. After 6 weeks, the number of caregivers still considering scratching to be a problem decreased from 99.2% to 58.7% for the group declaring scratching to be an issue and receiving FELIWAY Optimum, while the caregivers who declared an issue and did not use the product remained stable (97.7% to 95.4%).

The respondent-related aspect of validity was assessed using the question on caregivers’ consideration of scratching as a problem or not, and then compared with the index score value (Figure 2B). According to Figure 2B, approximately 90% of caregivers who did not consider scratching to have been an issue over the last 7 days had an index score value equal to 0. Almost all caregivers who considered scratching to have been an issue over the last 7 days had an index score value higher than 0.

The responsiveness of the scratching index score was evaluated based on categorisation of the cats according to the owner’s perception of the presence or absence of the problem at selection, as well as the intervention received (i.e., pheromone diffuser or nothing) (Figure 2C). The first observation that can be made is that the caregivers who declared scratching to be problem during selection were those with the higher mean index score. Secondly, the scratching index scores remained stable in the groups without any intervention, whereas in the group exposed to the pheromone diffuser at D14 the index scores decreased after this time, with a negative slope value over time (Figure 2C).

##### Urine Marking

First, the percentage of caregivers declaring that urine marking had been a problem during the previous 7 days is presented in Figure 3A. This demonstrates that at D0 the consideration of the problem was consistent with the reported incidence at selection, aside from the delay between the two questionnaires (1–2 weeks). This figure also shows that the caregivers’ declarations were consistent over 14 days when no product was applied. After 6 weeks, the number of caregivers still considering urine marking to be a problem decreased from 97.7% to 35.6% for the group declaring urine marking to be issue and receiving FELIWAY Optimum, while the caregivers who declared an issue and did not use the product remained stable (90.8% to 88.2%).

The respondent-related aspect of validity was assessed using the question on caregivers’ consideration of urine marking as a problem or not, and then compared with the index score value (Figure 3B). According to Figure 3B, approximately 96% of caregivers who did not consider urine marking to have been an issue over the last 7 days had an index score value equal to 0. Almost all caregivers who considered urine marking to have been an issue over the last 7 days had an index score value higher than 0.

The responsiveness of the urine marking index score was evaluated based on categorisation of the cats according to the owner’s perception of the presence or absence of the problem at selection, as well as the intervention received (i.e., pheromone diffuser or nothing) (Figure 3C). The first observation that can be made is that the caregivers who declared urine marking to be problem during selection were those with the higher mean index score. Secondly, the urine marking index scores remained stable in the groups without any intervention, whereas in the group exposed to the pheromone diffuser at D14 the index scores decreased after this time, with a negative slope value (−0.72) over time (Figure 3C).

##### Fear

First, the percentage of caregivers declaring that fear behaviour had been a problem during the previous 7 days is presented in Figure 4A. Here, it is possible to observe a drop between D0 and the previous consideration of the problem, but only for the caregivers with an issue who received FELIWAY Optimum. For the group with the issue and no product, the consideration of the problem at D0 was consistent with the reported incidence at selection, aside from the delay between the two questionnaires (1–2 weeks). After 6 weeks, the number of caregivers still considering fear to be a problem decreased from 78.2% to 43% for the group declaring a fear issue and receiving FELIWAY Optimum, while the caregivers who declared an issue and did not use the product remained stable (89.5% to 88.6%).

The respondent-related aspect of validity was assessed using the question on caregivers’ consideration of fear as a problem or not, and then compared with the index score values (Figure 4B). According to Figure 4B, approximately 87% of caregivers who did not consider fear to have been an issue over the last 7 days had an index score value equal to 0. Almost all caregivers who considered fear to have been an issue over the last 7 days had an index score value higher than 0.

The responsiveness of the fear index score was evaluated based on categorisation of the cats based on the owner’s perception of the presence or absence of the problem at selection, as well as the intervention received (i.e., pheromone diffuser or nothing) (Figure 4C). The first observation that can be made is that the caregivers who declared fear to be problem during selection were those with the higher mean index score. Secondly, the fear index scores remained stable in the groups without any intervention, whereas in the group exposed to the pheromone diffuser at D14 the index scores decreased after this time, with a negative slope value (−0.52) over time (Figure 4C).

##### Cohabitation

First, the percentage of caregivers declaring that cohabitation behaviour had been a problem during the previous 7 days is presented in Figure 5A. This demonstrates that at D0 the consideration of the problem was consistent with the reported incidence at selection, aside from the delay between the two questionnaires (1–2 weeks). This figure also shows that the caregivers’ declarations were consistent over 14 days when no product was applied. After 6 weeks, the number of caregivers still considering cohabitation to be a problem decreased from 98.8% to 54.3% for the group declaring cohabitation to be issue and receiving FELIWAY Optimum, while the caregivers who declared an issue and did not use the product remained stable (88.9% to 93.8%).

The respondent-related aspect of validity was assessed using the question on caregivers’ consideration of cohabitation as a problem or not, and then compared with the index score values (Figure 5B). According to Figure 5B, approximately 87% of caregivers who did not consider cohabitation to have been an issue over the last 7 days had an index score value equal to 0. Almost all caregivers who considered cohabitation to have been an issue over the last 7 days had an index score value higher than 0.

The responsiveness of the cohabitation index score was evaluated based on categorisation of the cats according to the owner’s perception of the presence or absence of the problem at selection, as well as the intervention received (i.e., pheromone diffuser or nothing) (Figure 5C). The first observation that can be made is that the caregivers who declared cohabitation to be problem during selection were those with the higher mean index score. Secondly, the cohabitation index scores remained stable in the groups without any intervention, whereas in the group exposed to the pheromone diffuser at D14 the index scores decreased after this time, with a negative slope value (−0.72) over time (Figure 5C).

## 4. Discussion

In this study, we assessed some aspects of the validity and reliability of the CABIAS using an index scoring system based on a combination of the frequency and perceived severity of four common problem behaviours in cats: scratching, fear, urine marking, and problems with inter-cat cohabitation. Based on our results, we suggest that this could represent a robust tool for future use in assessing and monitoring problematic feline behaviours. This index score is based on caregivers’ assessments of the frequency and severity of these problems, and the CABIAS has the potential to help veterinarians and behaviourists in collecting data on these problems in their patients at initial assessment, as well as being a tool for monitoring cats’ progress during therapeutic interventions. Additionally, the CABIAS could be a potentially useful research tool to evaluate the effectiveness of products or interventions in reducing problem behaviours. Further work exploring other aspects of reliability and validity, outside the scope of this study, will be valuable in further establishing the clinical and research usefulness of this CABIAS, but these initial results are encouraging.

Our findings demonstrated a high level of inter-rater reliability among caregivers within the same household. The consistently high ICC values between caregivers, along with the consistently good levels of agreement, indicated that the ratings provided by different individuals using the CABIAS were close in terms of absolute values. Notably, the index scores exhibited high inter-rater reliability on their first completion, without requiring any specific training prior to use or any experience in using the index score, underscoring its ease of use. These findings suggest that the index score might be able to be used by different individuals observing the same animal, while maintaining reliability. However, although the caregivers were asked not to discuss or communicate their scores with one another, we could not eliminate this possibility, and future studies should therefore look further at inter-rater reliability within and across different households where caregivers are completely blinded to one another.

The index scores used as the basis of this study were all derived from caregiver-reported data, with a semi-quantitative assessment of the frequency of problem behaviours and an owner impression of their intensity (or severity). Ideally, the study would have incorporated an objective and quantitative assessment of these aspects, e.g., by using continuous video recordings of the cats and their environment throughout the study. However, such measures introduce many challenges, like setting up sufficient cameras and angles to capture the required problem behaviour(s), and were beyond the scope of the current investigations. Although caregivers may not always have directly observed some of the problem behaviours (e.g., scratching and urination), so some data on frequency may lack accuracy, the use of a VAS for intensity assessments might partially mitigate this shortcoming. Once the questionnaires were submitted, the caregivers did not have any access to their answers, and so VAS scores for intensity were provided independently for each assessment. As already noted, behaviour scores should ideally be objective and robust and, albeit complex, future work could be undertaken to assess caregiver-reported aspects of frequency and intensity against true, objectively recorded data for further validation. Another possibility would be to determine the correlation between the CABIAS percentage change from baseline when a product is applied and an “overall assessment of efficacy” by both a clinician and caregiver, similar to the approach taken in the CADESI validation [23]. This methodology closely resembles the evaluation of effectiveness employed recently in clinical trials [24,25]. Consequently, it could be regarded as a subjective yet pertinent “gold standard” measure of treatment efficacy widely accepted in the field of veterinary behaviour.

Fear and anxiety are closely related emotions, yet they differ in their nature and response. Fear is a natural and immediate reaction to a real or perceived threat, triggering a fight-or-flight response. It is typically tied to a specific, identifiable stimulus and often subsides once the threat is gone. In contrast, anxiety is a more prolonged, diffuse, and often anticipatory feeling of unease or apprehension. It tends to lack a specific trigger and can persist even in the absence of an immediate threat [26,27]. In this article, we did not specify to the caregiver that they should record fear behaviour only when a trigger was present, so it is possible that some instances of anxiety were confused with fear. However, we specifically asked about the major causes of fear reactions (see Appendix B), and in the majority of cases the caregivers clearly identified a trigger (such as noise, encountering other animals, or someone or an animal visiting the household). Hence, it is possible to consider that most of the events recorded here were really related to fear.

Convergent validity was assessed by comparing the index scores with the question concerning the impact of the problem behaviour on the caregiver. This resulted in notably strong correlations indicating good convergence between the index score and the impact on the caregiver, as predicted. The perceived frequency and severity of the problem thus appear to relate strongly to the extent to which the problem has a negative impact on the caregiver. The CABIAS can therefore be used as a surrogate measure of the impact of the problem on the caregiver, with a higher index score suggesting an increased likelihood of caregiver distress. This is important, as caregiver concerns may lead to consequences for the welfare of the cat, including considerations of euthanasia or relinquishment.

We evaluated the responsiveness validity of the CABIAS by examining the changes in index scores in cats after the introduction of the pheromone diffuser (FELIWAY Optimum), compared with the changes in cats that were not exposed to the pheromone product. As expected [20], exposure to the pheromone diffuser resulted in a clear decrease in the index scores over time, which was estimated using slope calculations. This was in contrast to cats that were not exposed to the pheromone complex, where the index scores remained stable over the 6-week period. This demonstrates that the CABIAS is capable of tracking changes in owner-perceived or owner-reported problem behaviours in cats.

An important limitation of this study arises from the challenge of distinguishing between the actual decrease in problem behaviour(s) and potential misinterpretations by caregivers within the group using FELIWAY Optimum. This issue arises because both behavioural changes and misinterpretation/misreporting can occur and become intertwined when a product or a placebo is employed. The inability to differentiate between these reasons in this study poses a difficulty for fully assessing the validity of the index score’s responsiveness, and again, future studies incorporating objective and quantitative data will be required, ideally with blinding to the intervention and/or the use of a placebo product. However, the lack of a placebo in the current study allowed us to report on owner perceptions of the four problem behaviours over a six-week period in cats with no intervention applied (i.e., in the group not receiving the pheromone diffuser). To the best of our knowledge, no other study has attempted to document problem behaviours over this timescale without intervention.

We also found that the CABIAS effectively distinguished between the owner-reported presence or absence of a problem behaviour (respondent-related validation). An index score of 0 was linked to caregivers not perceiving the problem behaviour in more than 87–96% of cases. This highlights the capability of the index score to be a good indicator of whether a caregiver finds the behaviour to be an issue for them. Additionally, the consistency in responses regarding the presence or absence of problem behaviours was generally high between the selection questionnaire and day 0 (baseline) for all behaviours, except for fear in the group receiving FELIWAY Optimum at day 14. Notably, 20% of individuals who initially considered fear to be a problem changed their opinion between selection and D0, but not between D0 and D14 before Phase 2. The reason for this specific drop in reporting fear as a problem is not clear, but it did not impact the responsiveness results, since the percentage remained stable between day 0 and day 14, and the drop occurred only after the application of the product, as expected.

## 5. Conclusions

In this study, we described the CABIAS to measure four common problem behaviours in cats and evaluated certain aspects of its validity and reliability. Our results demonstrate that for the measures that we looked at, the CABIAS appears to be a robust tool for both assessing and monitoring problems with scratching, inter-cat cohabitation issues, urine marking, and fear. Furthermore, its simplicity and practicality make it a potentially valuable resource for veterinarians and behaviourists to monitor the effectiveness of interventions aimed at improving these problems. The CABIAS has the potential to promote a higher quality of life for these companion animals.

## Figures and Tables

**Figure 1 animals-13-02992-f001:**
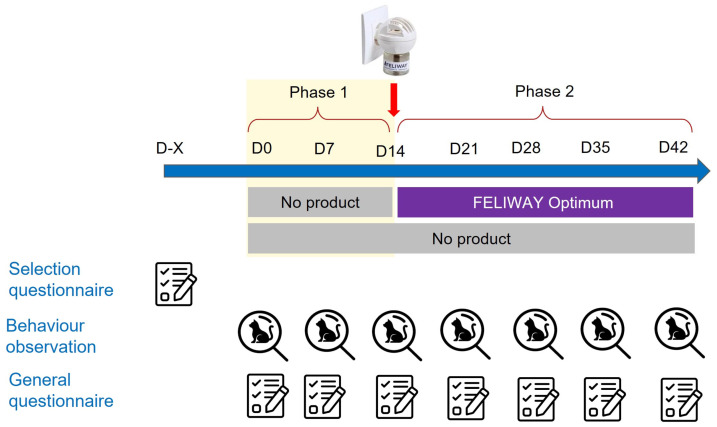
Scheme of the study protocol.

**Figure 2 animals-13-02992-f002:**
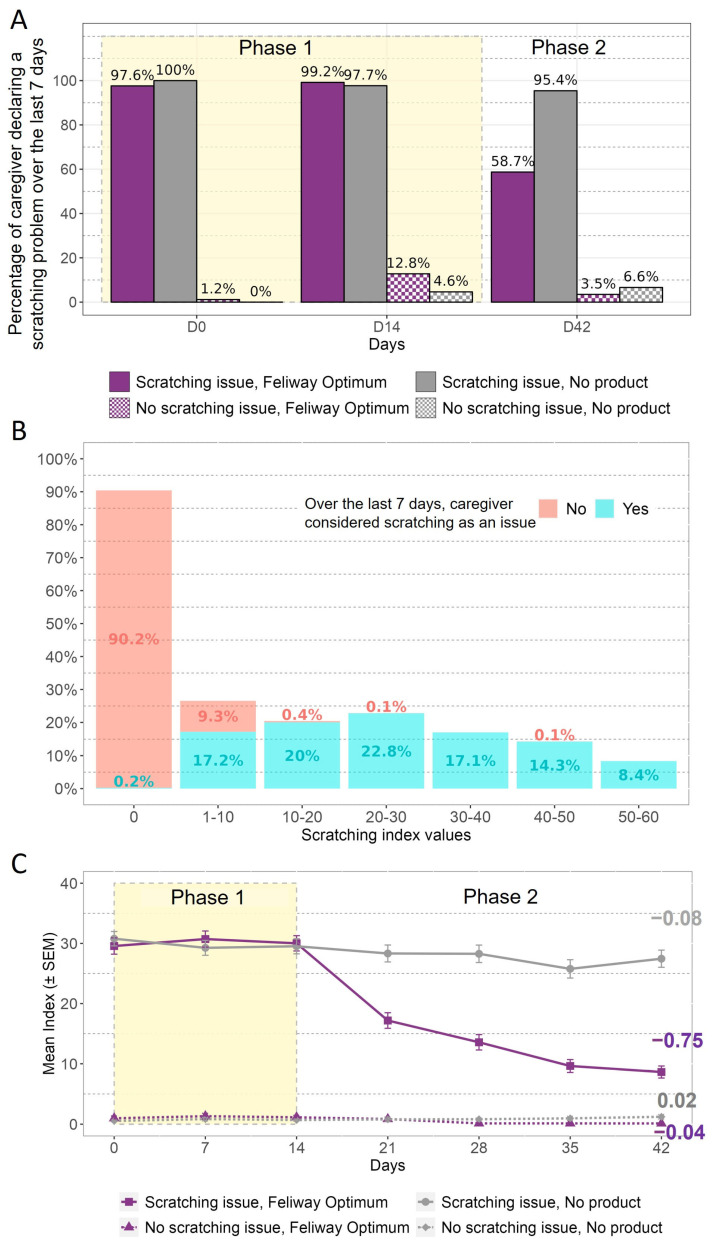
Scratching: (**A**) Percentage of caregivers considering scratching to have been an issue over the 7 last days, according to the day. (**B**) Distribution of the index score values according to whether the caregiver considered scratching to be an issue. (**C**) Mean index score evolution over the 6 weeks. The values on the graph represent the estimated slopes obtained using the linear mixed model of the index score change from D14 for each group. Phase 1 corresponds to the period where no product was applied, and Phase 2 is the period when some cats were exposed to the pheromone product.

**Figure 3 animals-13-02992-f003:**
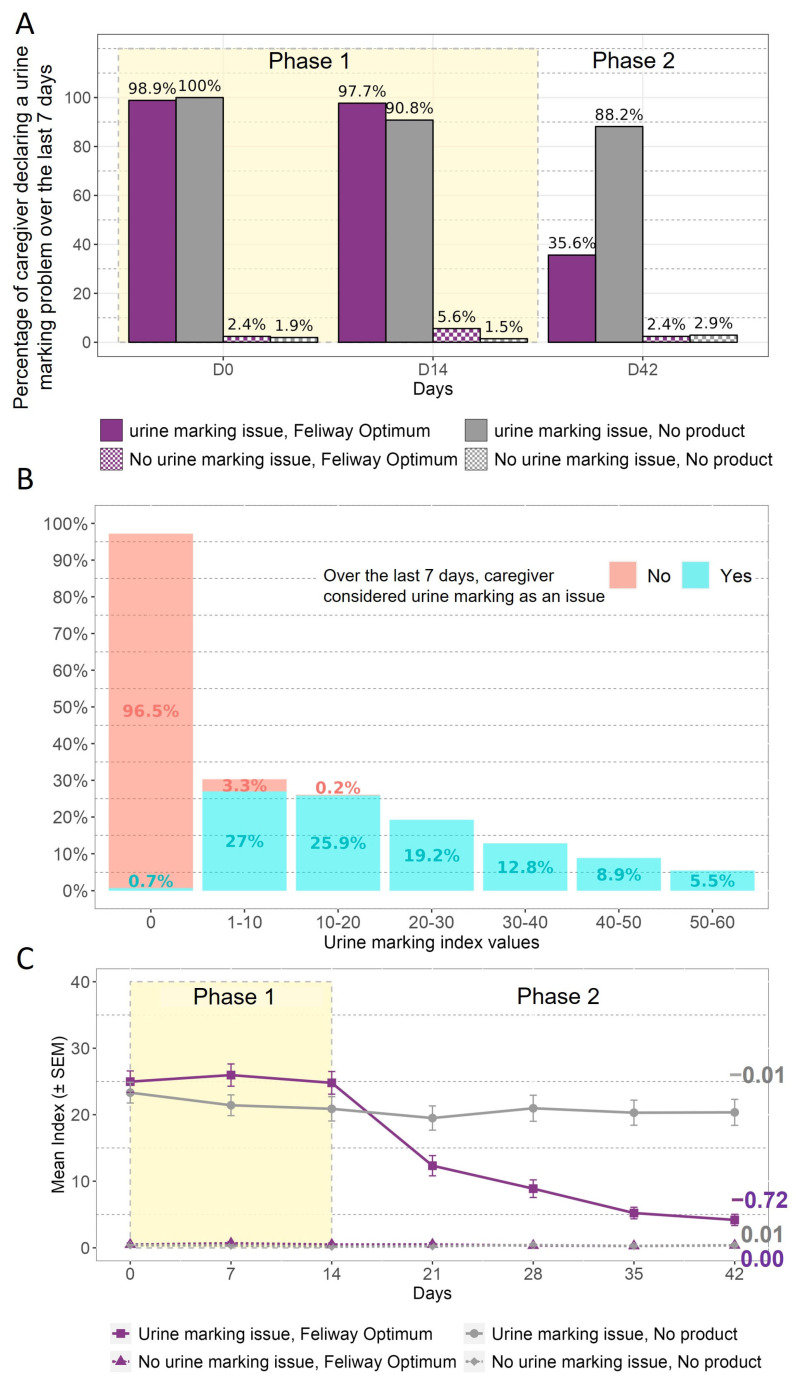
Urine marking: (**A**) Percentage of caregivers considering urine marking to have been an issue over the 7 last days, according to the day. (**B**) Distribution of the index score values according whether the caregiver considered urine marking to be an issue. (**C**) Mean urine marking index score evolution over the 6 weeks. The values on the graph represent the value of the slope of the estimated regression line of the change from D14 in each group. Phase 1 corresponds to the period where no product was applied, while Phase 2 is the period when the product was applied to some cats.

**Figure 4 animals-13-02992-f004:**
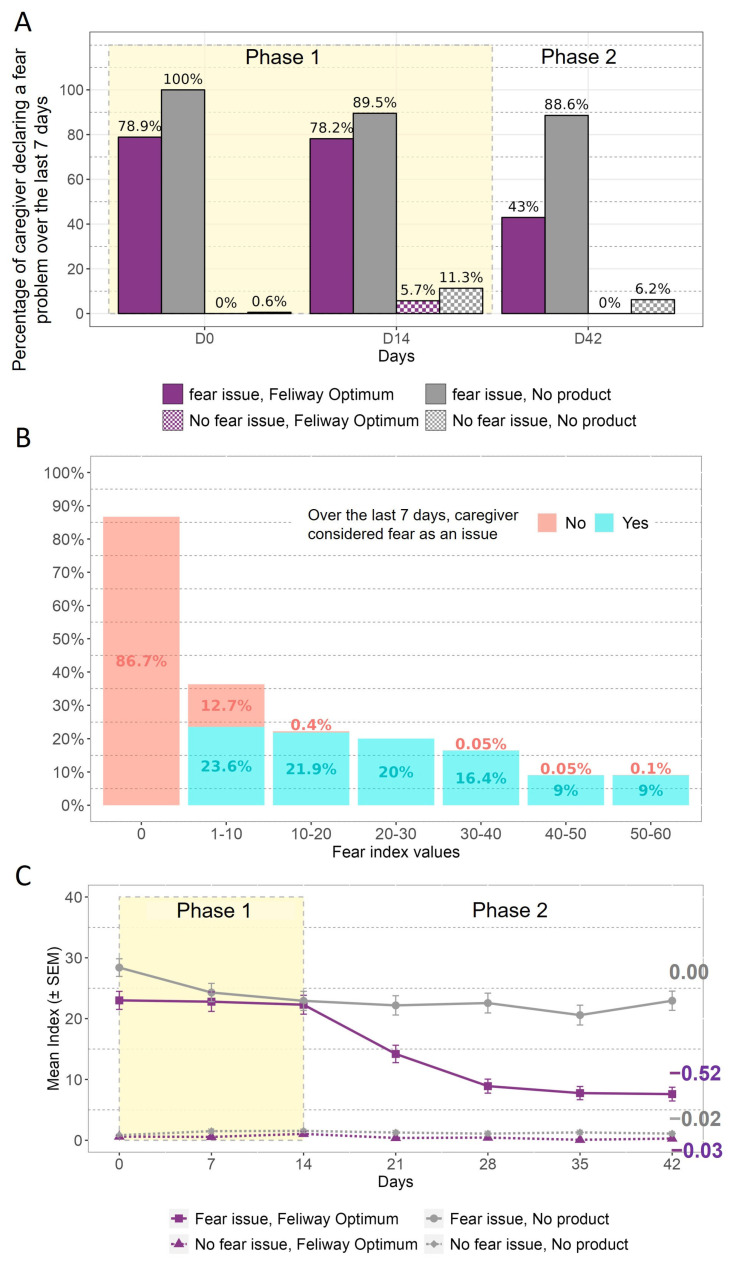
Fear: (**A**) Percentage of caregivers considering fear to have been an issue over the 7 last days, according to the day. (**B**) Distribution of the index score values according to whether the caregiver considered fear to be an issue. (**C**) Mean fear index score evolution over the 6 weeks. The values on the graph represent the value of the slope of the estimated regression line of the change from D14 for each group. Phase 1 corresponds to the period where no product was applied, and Phase 2 is the period where the product was applied to some cats.

**Figure 5 animals-13-02992-f005:**
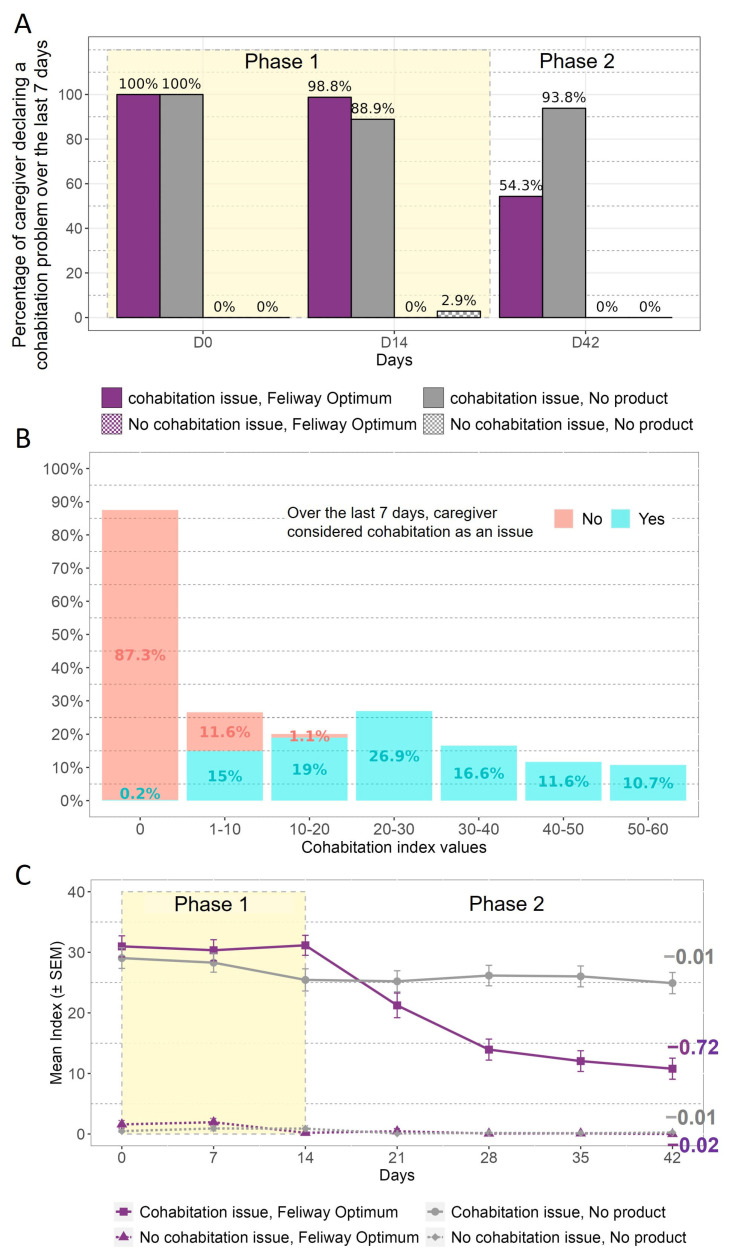
Cohabitation: (**A**) Percentage of caregivers considering cohabitation to have been an issue over the 7 last days, according to the day. (**B**) Distribution of the index score values according to whether the caregiver considered cohabitation to be an issue. (**C**) Mean cohabitation problem index score evolution over the 6 weeks. The values on the graph represent the value of the slope of the estimated regression line of the change from D14 for each group. Phase 1 corresponds to the period where no product was applied, while Phase 2 is the period where the product was applied to some cats.

**Table 1 animals-13-02992-t001:** List of exclusion criteria.

Exclusion Criteria	Description of Criteria
Cat age	Cats aged <12 months
Outdoor access	Cats spending >50% of their time outdoors
Cat’s housing	Cats kept away from their home during the previous month (e.g., hospitalisation, cattery, shelter, etc.)
Number of cats	Household with >2 cats
Cat’s health	Existing current disease diagnosed by a veterinarian
Cat’s environment	Household without a scratching post Household without a litter box
Other products	Current or recent (in the past month) use of pheromone or calming products (nutraceuticals, pharmaceuticals, environmental products, etc.)
Caregivers’ background	- Caregivers belonging to a research panel dedicated to a specific brand or a specific product manufacturer - Caregiver working in one of the following areas: the pharmaceutical industry, a veterinary practice or clinic, a pet store or shop, manufacture or distribution of pet food or pet care products, or a survey/market research institute

**Table 2 animals-13-02992-t002:** List of post-inclusion removal criteria.

Exclusion Criteria	Description of Criteria
Environmental changes	Major environmental change (examples available in Table 3)
Other products	Starting to use pheromone products or calming products (nutraceuticals, pharmaceuticals, environmental products, etc.)
Missing data	If ≥1 questionnaire was not answered, regardless of the reason
Observation not possible	Caregiver was not able to observe the cat for at least 4 h a day for a minimum of 5 days during every week of the study

**Table 3 animals-13-02992-t003:** Reasons for post-inclusion removal during the study.

Reasons for Exclusion	Number of Households
Lost during follow-up	33
Caregiver not present for the minimum amount of time	12
New pet in the household	7
Cat household change (house moving/vacation)	3
New scratching post introduction	1
Use of forbidden treatment	1
Cat welfare reason (child cut cat’s whiskers)	1
Cat sterilisation	1
New baby	1

**Table 4 animals-13-02992-t004:** Repartition of the population across the different groups and behaviours; groups with * contain 81 cats with no problem behaviour at all plus those that had not declared this behaviour as a problem; ** 17 cats had no problem at all with 2 cats in the household, plus those that did not declare a cohabitation issue but did declare another problem behaviours.

	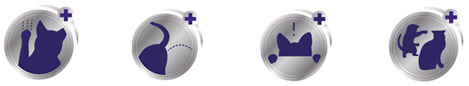
		Scratching	Urine Marking	Fear	Cohabitation
	N = Number of Cats	N = 494	N = 494	N = 494	N = 220
**Behaviour issue** **declared**	**Feliway Optimum**	126 (26%)	87 (18%)	142 (29%)	81 (37%)
**No product**	131 (27%)	76 (15%)	105 (21%)	81 (37%)
**No Behaviour issue** **Declared**	**Feliway Optimum**	86 (17%)	125 (25%)	70 (14%)	23 (10%)
**No product**	151 (31%) *	206 (42%) *	177 (36%) *	35 (16%) **

**Table 5 animals-13-02992-t005:** Inter-ater reliability between participants from the same household for each problem behaviour.

	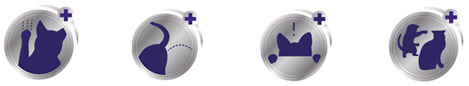
Scratching	Urine Marking	Fear	Cohabitation
ICC of Index (Lower−Upper)	N = 494	N = 494	N = 494	N = 220
D0	0.94 (0.92−0.95)	0.95 (0.94−0.96)	0.93 (0.92−0.94)	0.90 (0.86−0.92)
D7	0.93 (0.92−0.94)	0.96 (0.95−0.97)	0.93 (0.92−0.94)	0.91 (0.89−0.93)
D14	0.93 (0.92−0.94)	0.97 (0.97−0.98)	0.93 (0.91−0.94)	0.91 (0.88−0.93)
D21	0.94 (0.92−0.95)	0.96 (0.95−0.96)	0.94 (0.93−0.95)	0.88 (0.85−0.91)
D28	0.95 (0.94−0.96)	0.97 (0.96−0.97)	0.93 (0.92−0.94)	0.94 (0.92−0.95)
D35	0.95 (0.94−0.96)	0.97 (0.97−0.98)	0.93 (0.92−0.94)	0.93 (0.92−0.95)
D42	0.96 (0.95−0.96)	0.97 (0.96−0.98)	0.95 (0.94−0.95)	0.93 (0.91−0.95)
**Mean ± SD**	**0.94 ± 0.01**	**0.96 ± 0.01**	**0.93 ± 0.01**	**0.91 ± 0.02**

**Table 6 animals-13-02992-t006:** Percentage of inter-rater agreement ≤|10|.

	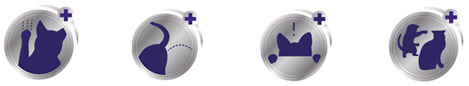
Scratching	Urine Marking	Fear	Cohabitation
Difference between Inter Rater in [−10;10]	N = 494	N = 494	N = 494	N = 220
**Globally**	**85.6%**	**94.6%**	**87.6%**	**79.9%**

**Table 7 animals-13-02992-t007:** Correlation between index score and impact for each behavioural issue.

	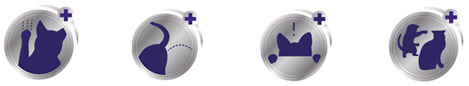
Index–Impact Correlation	Scratching	Urine Marking	Fear	Cohabitation
Pearson Coeficient	N = 494	N = 494	N = 494	N = 220
D0	0.91	0.86	0.89	0.88
D7	0.90	0.86	0.90	0.86
D14	0.92	0.88	0.87	0.89
D21	0.90	0.85	0.89	0.88
D28	0.91	0.86	0.87	0.90
D35	0.92	0.88	0.90	0.93
D42	0.92	0.90	0.91	0.92
**Mean ± SD**	**0.91 ± 0.01**	**0.87 ± 0.02**	**0.89 ± 0.02**	**0.86 ± 0.02**

## Data Availability

The data required to replicate all findings reported in this study are available from the corresponding author upon reasonable request.

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
