# Peer review of "Development of a Cat Behaviour Issues Assessment Scale (CABIAS) Assessing Problem Behaviours in Cats"

_animals, 2023, doi:10.3390/ani13182992_

Round 1
Reviewer 1 Report
Overall, this is a well written manuscript on a very important topic. The authors have done an excellent job with the study design and focusing on four primary behavior problems that are often cited as the reasons for relinquishment and euthanasia. Figure 1 showing the study timeline was well done and easy to understand. The inclusion and exclusion criteria tables were well constructed and easy to follow. Tables 4-7 were well constructed and the information was easy to read and understand.
My comments for clarifications or revisions are as follows:
Line 68: The word Scale, the “s” should be lower case
Line 79: Remove the words “So even in”. The sentence starts well with “Cats with multiple problem behaviors, the primary problem”
Line 157: Words are missing to complete a sentence. The sentence currently reads “The caregivers were also asked each week best scAdditionally, caregivers were asked….” Please complete the sentence with “best sc….”
Line 168 and Line 179- The subheadings under statistical analysis are not needed and make it hard to follow. Please remove.
Lines 194-198- This information is results, not materials/methods. Please move it to the next section.
Lines 200-203- The “Response Related Validation” is confusing. Can you please clarify this paragraph? What do you mean by the “Index score is discriminant”?
Page 10 of 29- Lines 287-289 (Figures do not have line numbers) Figure 2b- The presentation of this information is confusing. “Over the last 7 days, owner considered scratching as an issue- D0-90.2% “no”. I realize this in an Index score but it is hard to understand what you are presenting here that shows scratching is an issue. Can you present this data differently?
Page 12 of 29- Lines 320-326 (Figures do not have line numbers) Figure 3b- The presentation of this information is confusing. “Over the last 7 days, owner considered urine marking as an issue- D0-96.5% “no”. I realize this in an Index score but it is hard to understand what you are presenting here that shows urine marking is an issue. Can you present this data differently?
Page 14 of 29- Lines 355-361 (Figures do not have line numbers)
Figure 4a- Please clarify why, at D0, the caregivers in the no product group are 100% with considering fear an issue but only 78.9% of the caregivers in the group that received an intervention think fear is an issue. Shouldn’t both groups be in the high 90’s at the beginning?
Figure 4b- The presentation of this information is confusing. “Over the last 7 days, owner considered fear as an issue- D0-86.7% “no” and there is no % for yes. I realize this in an Index score but it is hard to understand what you are presenting here that shows fear is an issue. Can you present this data differently?
Page 16 of 29- Lines 390-397 (Figures do not have line numbers)- Figure 5b- The presentation of this information is confusing. “Over the last 7 days, owner considered cohabitation as an issue- D0-87.3% “no”. I realize this in an Index score but it is hard to understand what you are presenting here that shows cohabitation is an issue. Can you present this data differently?
Reviewer 2 Report
Congratulations on producing your manuscript. Feline behaviour is a topic dear to me and I appreciate the work involved.
Please take these comments below in the spirit that they were intended, namely to assist you in producing the best article possible. Best wishes.
1. Can you justify using Pearson correlation rather than Spearman? Although most statisticians are happy to use parametric statistics for sliding scales, your frequency scale is definitely ordinal. Might be worth a comment in your manuscript.
2. For each problem behaviour, there was an increase from D0 to D14 in the non reported groups. Do you think owners were paying more attention to their cat’s behaviours?
3. There was also often a decrease from D0 to D14 in the reported groups with no product. Do you think owners were consiously or subconsciously putting strategies in place?
4. Was the questionnaire in English or French? It appears to have been translated into English except for Yes/No. Also, did the first 5 questions deal with demographics?
5. Fig. A2, while correlation is high, the data is extremely spread out. E.g. urine marking, scores for impact of 10.0 range from about 3 to 60. How much confidence do you have in your line of best fit?
Minor comments:
Line 103: When you state “free of any problem behaviour” do you mean free of the 4 categories you consider? There are other problem behaviours not considered in this manuscript.
Line 157: “week best scAditionally” needs to be fixed.
Line 231: reference isn’t numbered as all other references are
Only a couple of minor suggestions:
Line 106: “The first caregivers register to the panel” should perhaps be “The first caregivers to register to the panel”
Line 332: “that will be received Feliway” should perhaps be “that will receive Feliway”
